# Are change of direction speed and agility different abilities from time and coordinative perspectives?

**Mónica Morral-Yepes**[1,2], **Oliver Gonzalo-Skok**[3‡]*, **Thomas Dos´Santos**[4‡], **Gerard Moras Feliu**[1]

**1** Department of Sports Performance, INEFC, Universitat de Barcelona, Barcelona, Spain, **2** Department of Health Sciences, Research Group in Technology Applied to High Performance and Health, Universitat Pompeu Fabra, Barcelona, Spain, **3** Department of Communication and Education, Universidad Loyola Andalucía, Sevilla, Spain, **4** Department of Sport and Exercise Sciences, Musculoskeletal Science and Sports Medicine Research Center, Manchester Metropolitan University, Manchester, United Kingdom

☯ These authors contributed equally to this work.
‡ OGS and TDS also contributed equally to this work.
* oligons@hotmail.com

**Data Availability Statement:** Data from the current study can be found in the following DOI: 10.6084/m9.figshare.24198516.

## Abstract

This study aimed to test whether agility and change of direction speed (COD) are independent capacities using the same movement pattern (1) in terms of the completion time and (2) the entropy. Seventeen semi-professional female football players participated in the study. The agility task consisted of a Y-shaped (45˚ COD) task with three possible exit options (center, right and left) performed pre-planned or in reaction to the movement of two testers (i.e., blocking exit gates). Players' acceleration was measured using an inertial measurement unit. Entropy was calculated from the acceleration signal and completion time was extracted using a magnet-based timing system. Significantly greater times and lower entropy ($p < 0.001$) were found during agility runs to pre-planned COD runs. Furthermore, weak to moderate correlations were found between COD and agility for both completion time ($r = 0.29$, $p < 0.001$) and entropy ($r = 0.53$, p<0.001, $r^2 = 28.1\%$). These results highlight that COD speed and agility are independent capacities and skills, and as such, should be tested and trained as distinct, separate qualities. Modifying task constraints including a reactive stimulus (i.e., cognitive factors), is essential for increasing task complexity by altering the biomechanical and coordinative aspects of the action.

## Introduction

Team sports are characterized by a combination of actions that includes constantly changing movement patterns and speed depending on the sport situation [1]. Due to their intrinsic nature, agility is a key performance indicator in multidirectional sports [2–4], as players must constantly change direction in response to various stimuli based on the game situation. Along the same line, researchers have shown that agility can distinguish between different age categories and performance levels [3–5]. Agility has been defined as a "rapid and accurate whole-

**Funding:** The author(s) received no specific funding for this work.

**Competing interests:** The authors have declared that no competing interests exist.

body movement with change of velocity, direction, or movement pattern in response to a stimulus" [6], involving perceptual-cognitive factors, and physical and technical factors related to change of direction (COD) speed [7]. As such, agility and COD speed have been differentiated into two concepts during the last few years. Agility is the ability to react/anticipate a stimulus and focuses on perceptive/cognitive factors [8]. On the other hand, COD speed is the ability to decelerate, reverse or change movement direction, and re-accelerate without reacting to an external stimulus [9–11]. Based on this differentiation in concept, there has recently been a growing number of studies dedicated to testing whether or not COD speed and agility are independent abilities and skills, finding a positive answer to this question in most cases (i.e. better COD speed does not necessarily equate to better agility performance, and vice versa) [12–14]. While insightful, the main limitation with previous research is that the conclusions were based solely on result variables as completion times, and the use of linear measures, based on the analysis of the data dispersion in relation to mean, such as standard deviation (SD) or the coefficient of variation (CV), for analysing the differences. This approach offered only quantitative information about the magnitude of the variation but overlooked the analysis of the execution and movement strategies, such as the coordination of the sporting action.

Notwithstanding the classical analysis of result variables provides relevant information about the outcome of the action, it remains incomplete as the underlying cause of the achieved result remains unknown. The use of non-linear techniques such as the entropy can fill this gap, as it takes into consideration the temporal structure and complexity of the data for the analysis [15]. This approach allows for understanding the evolution of human movement over time, emphasizing the exploratory nature of motion and facilitating the analysis of motor behaviour and its variations [16, 17]. Thus, the use of non-linear techniques has been determined as an excellent alternative to exploring the nature of human movement and its relationship with the coordinative aspects of action providing both quantitative and qualitative insights into the behaviour of the motor system [15, 18].

Within these non-linear techniques, entropy has been widely used for a qualitative analysis of different motor actions such as a serve in tennis, free throws in basketball, and a strength training task [19–23]. Entropy assesses the regularity of the signal so that a lower entropy value represents a higher regularity and predictability of the signal. Then, the probability of a similar pattern occurring is higher [24]. On the contrary, the more irregularity and unpredictability of the signal, the higher the entropy [25]. To the best of our knowledge, entropy has never been evaluated during COD tasks performed in either pre-planned or reactive situations. Thus, its use and analysis can help practitioners to better understand the underlaying factors influencing the execution of these actions, and to determine whether the well-established distinctions between the two capacities traditionally assessed through result variables such as completion time, are also found from a coordinative perspective when employing non-linear measures.

Therefore, this study aimed to establish whether COD speed and agility are independent capacities using the same movement pattern (1) in terms of a result variable as the completion time, and (2) in terms of a process variable as the entropy. We hypothesized that COD speed and agility are different capacities, considering the action's time [12, 14] and entropy [26, 27]. Specifically we expect finding longer completion times and lower entropy during agility tasks [27].

## Material and methods

### Participants

Seventeen highly-trained female football players (age: 19.6 [3.24] years; body mass: 57.5 [6.79] kg; height: 1.63 [0.06] m; 11.2 [3.67] years of football experience) belonging to the same professional football squad (i.e., Spanish 2$^{nd}$ Division "Reto Iberdrola") voluntarily took part in the

current study. This sample size ($n$ = 17) was selected to detect an intraclass correlation coefficient (ICC) of 0.9 at 80% power (https://wnarifin.github.io/ssc/ssicc.html) in the reliability analysis. It was also selected to detect a moderate difference (ES: 0.7) (15 participants were needed) for a paired sample t-test at 80% power and alpha of 0.05 according to G*power (version 3.1.9.6). All players agreed to participate in the study by signing an informed consent form. For players under 18 years of age, informed consent was obtained from their parents or legal guardians. Data collection took part at the beginning of November 2021, during the 12th microcycle of the season. All players participated, on average, in four two-hour-long football training sessions, and one competitive match per week. A typical training session included a first part of strength training in the gym lasting about 30 minutes and 1hr 30 minutes of technical-tactical work on the pitch. The procedures complied with the Declaration of Helsinki (2013) and were approved by the local ethics committee (005/CEICEGC/2021). All subjects gave their written informed consent before participating in the study.

## Design

A cross-sectional mixed research design was employed (within-subject comparative design). A group of highly-trained female football players were assessed during COD speed and a football agility task. All participants were familiarized with testing procedures before starting the experiments, as these tasks were commonly included during previous training sessions. The football agility task was designed based on the most typical agility test considerations [9]. It consisted of a Y-shaped course with three possible exit gates: straight, left, and right. The total distance of the test was 10 m, consisting of a first section of 4 m (entry) to the midpoint, and 6 m (exit) for the second section to the final of each exit gate. Participants completed either a 45˚ COD to the right or to the left, or straight ahead sprint. At the midpoint of the agility task, a visible circle of 1 m in diameter was made to mark the point where the player must perform a COD towards the final gate. Four timing gates were used (i.e., at the beginning, and after each of the three possible options) to measure completion time. Participants started the test at their discretion 1-m away from the first two magnet-timed gates and were instructed to complete the tasks as fast as possible, with a line placed 2-m ahead of the exit gates to avoid slowing down before reaching the final line (Fig 1).

The tasks were performed either with (i.e., agility) or without reacting to a stimulus (i.e., COD speed). The pre-planned situation consisted of knowing the movement direction, while the agility task involved reacting to a stimulus, making a decision, and executing a motor action. As players arrived at the middle-point, two researchers moved to two of the three final options, according to a previously scripted sequence of imposed constraints requiring the participant to run as fast as possible to the free exit gate (Fig 2). All these options resulted in six different runs (i.e., three pre-planned and three with reactions) and each player performed each run four times in a randomized order, resulting in a total of 12 pre-planned and 12 reactive trials per player.

The task was performed on the usual training pitch in the evening (6 to 8 PM). Players were instructed to follow their habitual pre-training routine. The protocol consisted of a standardized 10-minute warm-up consisting of running, dynamic joint mobility exercises and progressive runs. Afterwards, players were allowed to perform the task once for each running category to familiarize with it with a rest of 2 min before starting the assessment. During the testing, participants had a rest period of at least 60 s after each run. Once the test started, there was a rest period of 10–15 min after each 10 runs before proceeding with the next runs. The runs were randomly assigned to each player. All trials were recorded by video.

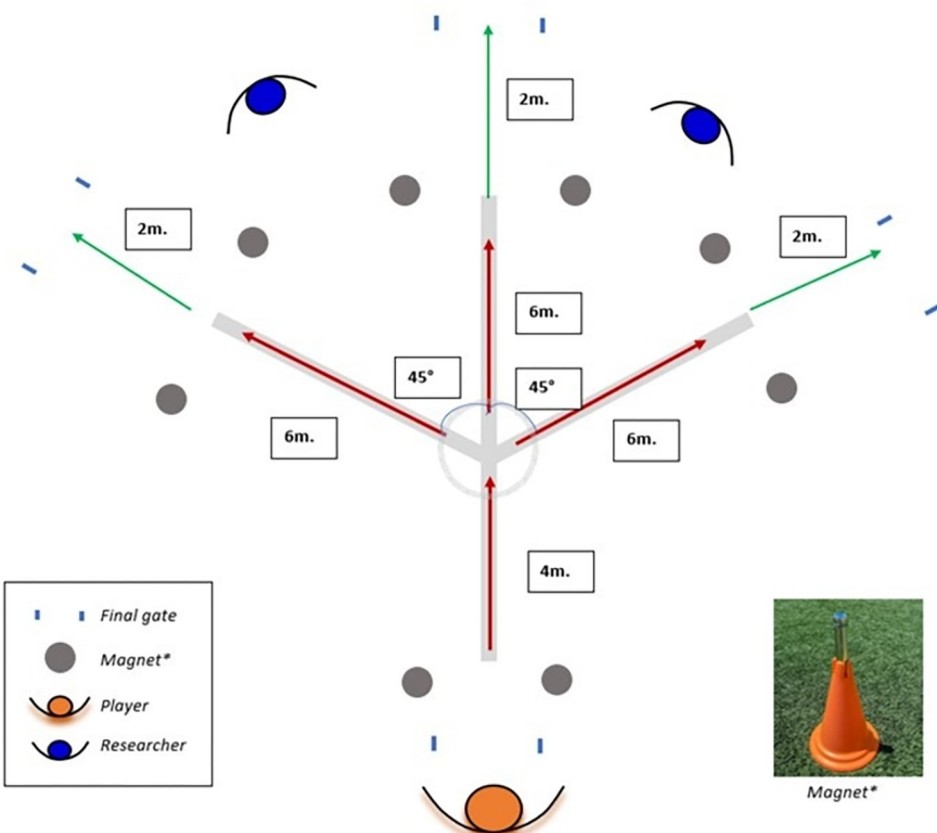

**Fig 1. Graphical representation of distances, angles, and structural characteristics of the football agility task performed.**

## Procedures

Players' acceleration was measured using an inertial measurement unit (*WIMU*, *Realtrack Systems*, *Almeria*, *Spain*), with a 3D accelerometer 100G, recording at 1000 Hz and a 3D magnetometer recording at 100 Hz. Such device was attached to players using an elastic waist belt close to the sacrum with a hard fixation to avoid extraneous acceleration during running [28].

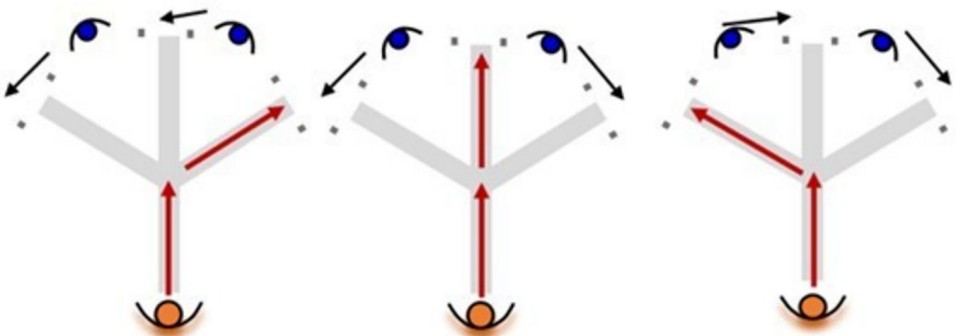

**Fig 2. The three reaction options based on the movement of the two testers in the agility task.**

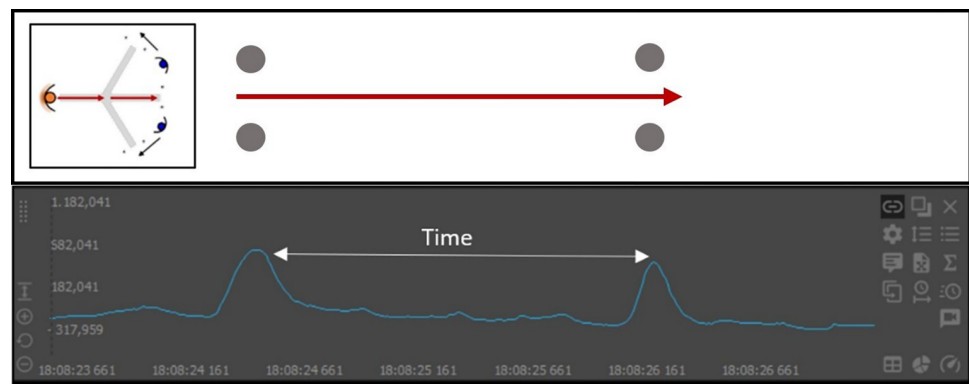

**Fig 3. Detected peak in the magnetometer of the IMU when crossing the gate with the magnets used to determine the time of each run.**

Two magnets (D33 × 267 mm, ND35, A.C. magnets 98, Barcelona, Spain) were placed at a height of 450mm above the ground on each gate of the task following the magnet-based timing system methodology purposed by Pérez-Chirinos et al. [29] to indicate the time of passage from the player through each gate. With this method, as the player crossed the gate with the magnets, the magnetometer of the IMU increased the signal generating a detectable peak and allowing to cut the signal through these peaks to determine the time of each run (Fig 3).

A portable high-speed camera (*Casio Exilim EX-ZR100*) recording at 240 fps was placed perpendicular to the exit gates of the test to ensure that each run corresponds to the one marked on the note paper, and thus avoiding possible errors in the interpretation of the signal.

## Data analysis

Each time a football player passed near to a bar magnet, a peak in the magnetometer time series was detected (Fig 3). SPRO software (*Realtrack Systems*, *Almeria*, *Spain*) was used to analyse the data recorded with the IMU to calculate the elapsed time between peaks. The magnetometer registered the peaks corresponding to the time of passage through the initial and final gate of the test, permitting the cutting of acceleration signal corresponding to each trial to obtain the time spent in each run [29]. The devices were calibrated prior to their placement. This was done with a self-calibration system that incorporates each device in the internal configuration of the boot. During self-calibration, three aspects were taken into account: (i) leaving the device immobile for 30 s; (ii) placing it in a flat area; and (iii) ensuring no other magnetic devices were around it [30]. This device has reported good results in both accuracy and reliability of its different sensors in previous studies [30–34]. The raw acceleration signal was extracted from each device and processed using a summation of vectors (AcelT) in three axes, mediolateral (x), anteroposterior (y) and vertical (z) calculated according to Gómez-Carmona et al. (2018).

The Sample entropy (SampEn) for each accelerometry signal, corresponding to each of the runs recorded through the IMU and extracted to an Excel file, was calculated using the equation proposed by Richmann & Moorman (2000) [35]. Then, dedicated routines programmed in Matlab® (*The MathWorks*, *Massachusetts*, *USA*) were used with a template length m of 2, and the tolerance criterion of 0.20 in the analyses.

## Statistical analysis

Data were processed using the SPSS software package (*Version 25; SPSS*, *Chicago*, *IL*, *USA*) and Microsoft Excel (*Version 2212*, *Microsoft*, *Redmond*, *WA*, *USA*). Basic descriptive statistics

(mean, standard deviation, minimum, maximum) were calculated for all measures. As the sample of the test was >50, the normality of the data was examined using Kolmogorov-Smirnov, finding that completion time variables did not fit a normal distribution ($p<0.05$), while the entropy variables did ($p>0.05$). Therefore, parametric tests were used to analyse the entropy variables and non-parametric tests were used for the time variables.

Within-session reliability analysis was computed using a 2-way random ICC with an absolute agreement and 90% confidence intervals and the typical error of measurement (TEM). Furthermore, the TEM was also expressed as the coefficient of variation (CV). The interpretation of intraclass correlation coefficient values aligned with the approach presented by Koo and Li [36], where values >0.9 = excellent, >0.75 to 0.9 = good, >0.5 to 0.75 = moderate, and <0.5 = poor; coefficient of variation values was considered as good (<5%), moderate (5–10%) and poor (>10%). To assess the difference between both conditions, a paired t-test was used when normality of the data was confirmed, otherwise Wilcoxon test was used. Also, a repeated measures ANOVA for entropy data, and a Friedman test for time data were performed to analyze the differences between each type of run, including a pairwise post-hoc comparison using Tukey and Durbin-Conover, respectively. The level of significance was set at $p<0.05$, with all calculations based on a 95% confidence interval (CI). Additionally, effect sizes (ES) were calculated to assess the magnitude of the difference between agility and COD, and between each type of run using Cohen's *d*. ESs were interpreted as follows: trivial (<0.20), small (0.20–0.49), moderate (0.50–0.79) and large (>0.80) [37].

The correlation analysis between COD and agility performance was also analyzed using Pearson product moment following a normal distribution, and Spearman test when followed a non-normal distribution. Results were interpreted as follows: 0 (zero), 0.01 to 0.39 (weak), 0.40 to 0.69 (moderate), 0.70 to 0.99 (strong) and 1 (perfect) [38].

## Results

Reliability results are shown in Table 1.

Significantly greater times ($p<0.001$, ES = 1.20,1.63) were found during agility tasks compared to COD speed tasks. Furthermore, a significantly lower entropy ($p<0.05$, ES = -0.64, -0.25) was observed during agility tasks compared to COD speed task.

The Friedman test used to assess the differences in time between the different exit options revels a significant effect ($\chi^2$ = 226, gl = 5, p< .001). Subsequent Durbin-Conover post-hoc test found significant differences ($p<0.05$) between all pair of options, except for two between COD Middle vs COD Right and between AG Right vs AG Left (Table 2).

**Table 1. Descriptives and reliability measures for COD speed and agility tasks.**

|  | Trial 1 | Trial 2 | ICC (90%CI) | TEM (90%CI) | CV (90%CI) |
|---|---|---|---|---|---|
| COD Middle | 1.72 ± 0.05 | 1.73 ± 0.06 | 0.91 (0.80, 0.96) | 0.02 (0.01, 0.03) | 1.1 (0.8, 1.5) |
| COD Right | 1.71 ± 0.07 | 1.72 ± 0.06 | 0.97 (0.92, 0.99) | 0.01 (0.01, 0.02) | 0.7 (0.6, 1.1) |
| COD Left | 1.72 ± 0.05 | 1.73 ± 0.06 | 0.99 (0.97, 0.99) | 0.01 (0.01, 0.01) | 0.6 (0.5, 0.9) |
| AG Middle | 1.79 ± 0.09 | 1.79 ± 0.10 | 0.96 (0.92; 0.98) | 0.02 (0.01, 0.03) | 1.1 (0.8, 1.5) |
| AG Right | 1.94 ± 0.11 | 1.95 ± 0.11 | 0.90 (0.78, 0.96) | 0.04 (0.03, 0.05) | 1.9 (1.5, 2.8) |
| AG Left | 1.94 ± 0.17 | 1.93 ± 0.15 | 0.96 (0.92, 0.98) | 0.03 (0.03, 0.05) | 1.6 (1.3, 2.3) |

CI: confidence intervals; COD: Change of direction (i.e. pre-planned); AG: agility (i.e. non-planned); ICC: Intraclass correlation coefficient; TEM: typical error of measurement; CV: coefficient of variation expressed as percentage of TEM.

The descriptive statistics of time and entropy from COD and agility runs are defined in the Fig 4.

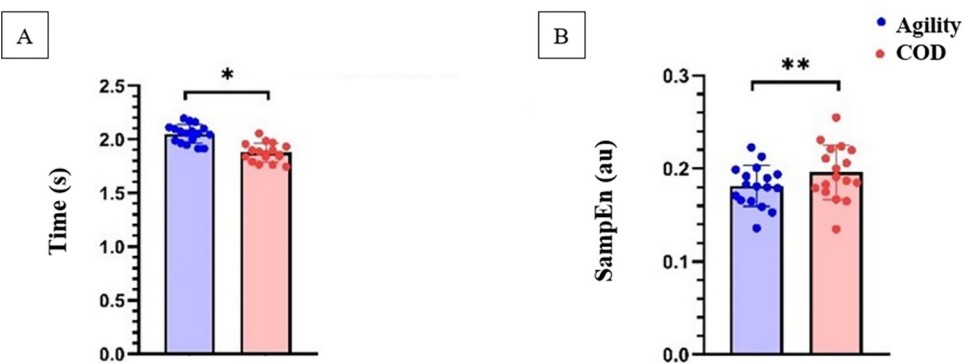

**Fig 4.** Descriptive statistics comparing A) time in seconds (s) and B) Sample Entropy in arbitrary units (au) between COD and agility runs. *p<0.01; **p<0.05.

The repeated measures ANOVA performed for entropy variables also revealed a significant main effect (F = 8.26, gl = 5, p<0.001). Post-hoc tests using Tukey's method showed significant differences (p<0.05) between six pairs of running options: COD Middle vs RA Left; COD Right vs AG Right/Left; COD Left vs AG Right/Left; AG Middle vs AG Left (Table 3).

Large increases (ES = 1.40 [1.20; 1.63]) in completion variables (i.e., time) and a small decrease (ES = -0.44 [-0.64; -0.25]) in entropy variables were observed when comparing agility tasks against COD speed tasks.

Weak to moderate correlations were found between COD speed and agility task completion times ($r = 0.29$, $p<0.001$) and moderate correlations in entropy ($r = 0.53$, $p<0.001$, $r^2 = 28.1\%$) (Fig 5).

## Discussion

The study aimed to test whether agility and COD speed are independent capacities using the same movement pattern (1) in terms of completion time and (2) entropy. To our best

**Table 2. Pairwise comparison results using Durbin-Conover post-hoc test for completion time.**

| Time | Statistic | p | ES | 95%CI |
|---|---|---|---|---|
| COD Middle–COD Right | 1.42 | 0.16 | -0.14 | [-0.48, 0.20] |
| COD Middle–COD Left | 3.47 | < .001 | -0.30 | [-0.64, 0.03] |
| COD Middle–AG Middle | 6.71 | < .001 | 0.69 | [0.35, 1.04] |
| COD Middle–AG Right | 14.49 | < .001 | 1.55 | [1.17, 1.93] |
| COD Middle–AG Left | 14.96 | < .001 | 1.76 | [1.36, 2.15] |
| COD Right–COD Left | 2.05 | 0.04 | -0.20 | [-0.54, 0.14] |
| COD Right–AG Middle | 8.13 | < .001 | 0.91 | [0.56, 1.27] |
| COD Right–AG Right | 15.91 | < .001 | 1.74 | [1.35, 2.14] |
| COD Right–AG Left | 16.38 | < .001 | 1.99 | [1.58, 2.40] |
| COD Left–AG Middle | 10.19 | < .001 | 1.02 | [0.67, 1.38] |
| COD Left–AG Right | 17.96 | < .001 | 1.81 | [1.41, 2.21] |
| COD Left–AG Left | 18.44 | < .001 | 2.04 | [1.63, 2.46] |
| AG Middle–AG Right | 7.78 | < .001 | 1.01 | [0.66, 1.37] |
| AG Middle–AG Left | 8.25 | < .001 | 1.19 | [0.82, 1.55] |
| AG Right–AG Left | 0.47 | 0.64 | 0.10 | [-0.24, 0.43] |

CI: Confidence intervals; COD: change of direction; ES: effect size; AG: agility

**Table 3. Pairwise comparison results using Tukey's post-hoc test for entropy.**

| Entropy | | Means difference | t | $p_{tukey}$ | ES | 95%CI |
|---|---|---|---|---|---|---|
| COD Middle | COD Right | 0.00 | -0.88 | 0.95 | 0.08 | [-0.25, 0.42] |
| | COD Left | 0.00 | -0.61 | 0.99 | 0.04 | [-0.29, 0.38] |
| | AG Middle | 0.00 | 0.64 | 0.99 | -0.07 | [-0.40, 0.27] |
| | AG Right | 0.01 | 2.47 | 0.15 | -0.32 | [-0.66, 0.02] |
| | AG Left | 0.02 | 3.79 | 0.00 | -0.49 | [-0.84, -0.15] |
| COD Right | COD Left | 0.00 | 0.40 | 1.00 | -0.04 | [-0.38, 0.29] |
| | AG Middle | 0.01 | 1.60 | 0.61 | -0.16 | [-0.50, 0.17] |
| | AG Right | 0.02 | 3.80 | 0.00 | -0.41 | [-0.75, -0.07] |
| | AG Left | 0.02 | 4.56 | < .001 | -0.60 | [-0.94, -0.25] |
| COD Left | AG Middle | 0.01 | 1.21 | 0.83 | -0.12 | [-0.45, 0.22] |
| | AG Right | 0.02 | 3.02 | 0.04 | -0.37 | [-0.71, -0.03] |
| | AG Left | 0.02 | 4.23 | < .001 | -0.56 | [-0.90, -0.21] |
| AG Middle | AG Right | 0.01 | 2.38 | 0.18 | -0.28 | [-0.62, 0.06] |
| | AG Left | 0.02 | 4.84 | < .001 | -0.48 | [-0.82, -0.14] |
| AG Right | AG Left | 0.01 | 1.41 | 0.72 | -0.18 | [-0.52, 0.15] |

CI: confidence intervals; COD: change of direction; ES: effect size; AG: agility; t: t-value.

knowledge, this is the first study assessing the differences and relationships between agility and COD speed by the result variable of the action (completion time), and by a process variable, as the movement variability of the action (entropy). Confirming our hypothesis, we found significant differences between performing agility and COD speed for both completion time and entropy. The main findings were a lower movement variability and a longer completion time when performing an unanticipated COD (i.e, agility) compared to pre-planned COD. Furthermore, the weak correlations between agility and COD speed indicate that these are independent capacities and skills, and highlight that athletes with superior COD speed will not necessarily demonstrate superior agility performance and vice versa.

Regarding completion time, the current findings align with previous studies highlighting the distinction between agility and COD speed [8, 13, 39]. Reinforcing this distinction, we also found significant differences between all type of tasks when comparing COD speed and agility.

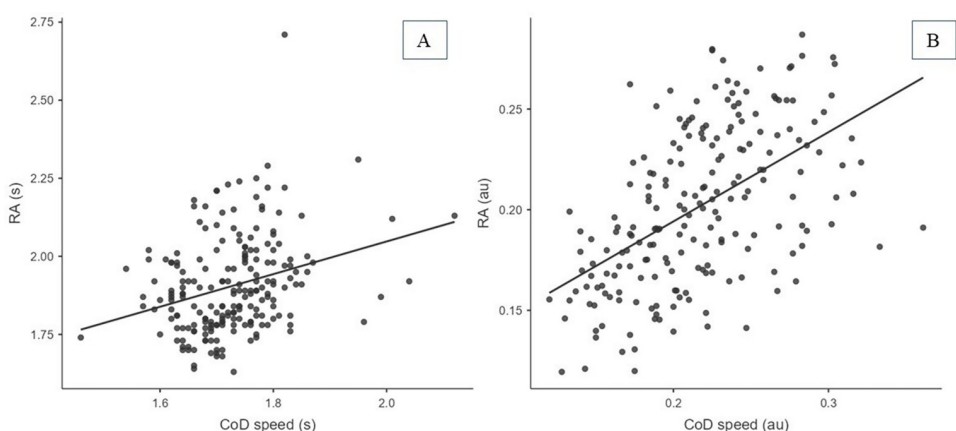

**Fig 5.** Correlation analysis between COD speed and agility (AG) values in A) Time (s) seconds, and B) Entropy variables (au) arbitrary units of Sample Entropy.

Moreover, some studies have found that agility can distinguish between different level players while COD speed does not [4, 40]. Conversely, other studies have found that COD speed is a critical ability to distinguish between performance levels [41, 42]. The discrepancy in the results could be attributed to the inclusion of a light stimulus in those studies or the inclusion of a relatively simple decision with the reaction to one stimulus with only two possibilities to go, which is quite different from the sport-specific stimulus where a player must react during a match. In this regard, it has been shown that performance during agility tasks using flashing lights/arrows does not differentiate skilful performers [14, 43]. Additionally, when athletes engage in these unplanned tasks in response to a flashing light or arrow, they still know that they will perform a time-constrained decision-making task in response to a stimulus in a laboratory environment [44]. In this context agility tasks containing generic stimuli such as flashing lights and arrows, although argued to provide a more challenging scenario compared to pre-planned tasks and having demonstrated their potential for enhancing athletic performance with their training and testing [45, 46], still lack ecological validity. These tasks do not provide the representative environments and the complex stimuli (i.e., cognitive loads) associated with the dynamic sport [44] as most of the tests include only one COD with two possibilities to go (right or left), minimizing the contribution of the cognitive factors of the action. Despite our study also including only one COD, it was in response to a sport-specific stimulus (i.e., 3D human stimulus), as the movement of two testers, and there were three exit possibilities (center, right and left), having more information to process and potential decisions to make, and conforming a more complex and specific task for the player. In this sense, it has been stated that experienced players react faster to sport-specific stimuli [8, 47]. Thus, those tests, including specific stimuli of sporting situations, have more ecological validity in assessing agility [8].

We also found significant differences in entropy between COD speed and agility tasks, with non-planned runs showing less entropy than pre-planned runs. Arguably, the inclusion of a constraint, such as the reaction to a stimulus, reduces the degrees of freedom of the player's sensorimotor system. Consequently, movement possibilities are constrained by the need to control the external variability imposed by the environment requiring the athlete to employ more controlled motor, cognitive, and sensory skills (thus contributing to a longer completion time) than movements done in a pre-planned manner, in an attempt to reduce the complexity of the task and control the movement [5]. To the best of our knowledge, no other studies have investigated the influence of including the reaction to a stimulus on movement variability (i.e., entropy) during COD motor tasks. However, previous research has examined the impact of including different constraints during tasks that require increased complexity or difficulty [48–50]. These investigations have demonstrated that the introduction of constraints can decrease movement variability among athletes as the task becomes more challenging. Specifically, these studies have identified that as the task's complexity increases with the inclusion of constraints, the structure of the signal becomes less intricate, resulting in a reduction in the athlete's movement variability [48–52]. It is possible that the inclusion of decision-making similarly contributes to the reduction of movement variability because of the increase in the difficulty of the action. However, further research is needed to fully understand the potential benefits of using constraints, and specifically decision-making, in motor skill development; and to be able to have reference values of what would represent a very high or very low movement variability and its influence on performance action. Generally, too low levels of variability (sub-optimal movement variability) have been associated with a very rigid and immutable system, and too much variability or a level of variability above the optimum as a chaotic and unpredictable system, in both cases being related to less adaptable systems [53].

We have also studied the differences between each type of run (center, right, and left) based on completion time and entropy, finding significant differences in general reactive and pre-

planned conditions. In the case of time, significant differences were found between all pre-planned runs compared to agility tasks. In the case of entropy, no significant differences were found when comparing the run performed through the centre with or without decision making, but there were differences between right and left exit COD options in the comparison between both conditions. Analysing why our study and many other have found no meaningful correlation, and significant differences between agility and COD speed tests, it's essential to recognise that while pre-planned COD speed tasks are primarily influenced by anthropometric, technical, mechanical, physical and motor capacities [54], in agility tasks the perceptual and decision-making skills as reaction time, visual scanning, anticipation, pattern recognition, and the knowledge of situations are critical components, playing a determining role in performance [5, 39, 55]. Thus, an athlete despite having good COD speed and action capacity may demonstrate sub-optimal decision-making time, speed, and accuracy, which negatively impacts overall agility completion time. Furthermore, the introduction of a stimulus can alter biomechanical movement, coordination, and muscle activation strategies when changing direction, with researchers indicating that unplanned sidestepping techniques are significantly different when compared to a pre-planned COD task [8, 56, 57]. Along the same line, some studies have found that even the angle of the change of direction [58], the type of stimulus [59, 60] and the timing of stimulus [61] can impact COD biomechanics.

The current study had a few limitations to consider when interpreting the results. Firstly, although using a human as the stimulus to respond may have higher ecological validity than generic light or arrow stimuli, as it more closely mimics real-world scenarios, it presents the inconvenience that the timing of the stimulus appearance is not fixed, as it depends on the movements of the testers. Also, it could be interesting to examine the use of different standardised sport-specific stimuli, such as reacting to a pass / opponent or including constraints that could enhance complexity of the agility task like a dribbling. Secondly, the investigation was limited to only one angle from a short entry distance, and the biomechanical demands of COD are angles and velocity dependent [58], and thus agility tasks of different angle and approach distances warrant further investigation. Additionally, the study only included female football players of a particular skill level and had a restricted sample size although sufficiently powered. Future research should aim to investigate gender comparisons, explore different skill levels, and consider expanding the sample size to attain more representative results. Lastly, the decision-time of the agility task understood as the time between the stimulus presentation and the initiation of the response, was not examined, which could provide valuable insights into perceptual-cognitive abilities and thus is a recommended area for future research.

## Conclusion

In conclusion, the results of this study found significant differences, and weak to moderate correlations, between CoD speed and agility in terms of completion time and entropy, highlighting that these are two distinct, independent qualities which should be assessed and trained as such. Also, modifying task constraints is crucial to enhance athlete's movement possibilities. Specifically, incorporating cognitive factors into a COD tasks can make it more sport-specific, potentially influencing the biomechanical and coordinative aspects of the action. In this regard, it could be interesting to initiate in a structured training program by focusing on the development of COD speed and gradually increasing complexity with the inclusion of different constraints, such as the reaction to a stimulus. This progression in the level of difficulty by the inclusion of cognitive constraints would induce changes at both coordinative and cognitive levels requiring new states of adaptation to the athlete.

## Acknowledgments

The authors would like to express their thanks to the participants for their enthusiasm and cooperation during the study.

## Author Contributions

**Conceptualization:** Mónica Morral-Yepes, Oliver Gonzalo-Skok, Gerard Moras Feliu.

**Data curation:** Mónica Morral-Yepes, Oliver Gonzalo-Skok, Thomas Dos´Santos, Gerard Moras Feliu.

**Formal analysis:** Mónica Morral-Yepes.

**Methodology:** Mónica Morral-Yepes, Oliver Gonzalo-Skok, Thomas Dos´Santos, Gerard Moras Feliu.

**Resources:** Mónica Morral-Yepes.

**Software:** Mónica Morral-Yepes.

**Supervision:** Mónica Morral-Yepes, Gerard Moras Feliu.

**Validation:** Mónica Morral-Yepes.

**Visualization:** Mónica Morral-Yepes.

**Writing – original draft:** Mónica Morral-Yepes, Gerard Moras Feliu.

**Writing – review & editing:** Mónica Morral-Yepes, Oliver Gonzalo-Skok, Thomas Dos ´Santos, Gerard Moras Feliu.

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
