## [Decision Letter · Decision Letter 0]

7 Nov 2023

PONE-D-23-32997Are change of direction speed and reactive agility different abilities from time and coordinative perspectives?PLOS ONE

Dear Dr. Gonzalo-Skok,

Thank you for submitting your manuscript to PLOS ONE. After careful consideration, we feel that it has merit but does not fully meet PLOS ONE’s publication criteria as it currently stands. Therefore, we invite you to submit a revised version of the manuscript that addresses the points raised during the review process.

We look forward to receiving your revised manuscript.

Kind regards,

Ersan Arslan, Ph.D.

Academic Editor

PLOS ONE

Journal Requirements:

2. Thank you for stating the following in your Competing Interests section: "No"

Reviewers' comments:

Reviewer's Responses to Questions

**Comments to the Author**

1. Is the manuscript technically sound, and do the data support the conclusions?

Reviewer #1: Partly

Reviewer #2: Yes

Reviewer #3: Yes

2. Has the statistical analysis been performed appropriately and rigorously? 

Reviewer #1: I Don't Know

Reviewer #2: Yes

Reviewer #3: Yes

3. Have the authors made all data underlying the findings in their manuscript fully available?

Reviewer #1: No

Reviewer #2: No

Reviewer #3: No

4. Is the manuscript presented in an intelligible fashion and written in standard English?

Reviewer #1: Yes

Reviewer #2: Yes

Reviewer #3: Yes

5. Review Comments to the Author

Reviewer #1: The study aimed to examine the independence of reactive agility (RA) and change of direction speed (CoD) utilizing the same movement pattern. Basic agility assessment metrics included completion time and entropy. The authors applied an ecological validity approach in measuring agility. While the paper is well-prepared, the chosen topic lacks novelty, given prior studies establishing the independence of CoD and RA (e.g., see Young, W. B., Dawson, B., & Henry, G. J. (2015). Agility and Change-of-Direction Speed are Independent Skills: Implications for Training for Agility in Invasion Sports. International Journal of Sports Science & Coaching, 10(1), 159–169). Moreover, kinematic distinctions in knee joint mechanics during running have been noted between RA and CoD tasks (Spiteri et al., 2015a; Thomas et al., 2020). Additionally, CoD plays a mediating role between sports discipline and RA (Domaradzki et al., 2021). The rationale for the study lacks persuasiveness in light of these earlier findings. If controversies exist concerning the CoD and RA relationship, the authors should emphasize them in the introduction, providing a clear justification. It is crucial to emphasize the new contributions to the existing literature, especially considering the established relationship between CoDs and RA, which is also confirmed under ecological validity conditions (Spiteri et al.)

The introduction requires further development. Some terms are poorly explained and challenging to follow. The literature review is inadequate.

• Line 64: "growing number of studies" – specify the studies for clarity.

• Lines 71–73: The sentence is unclear. Provide sources and rationale.

• Lines 88–89: The hypothesis formulated in the paper should align with prior literature.

• Line 92: I suggest using the term “Participants” instead of “Subjects”.

• Line 97: The choice of a large effect size (0.9) for sample size calculation seems questionable. A sample size of 17 appears small to ensure measurement reliability.

• Lines 126, 138 – The titles of the figures should be more specific.

The discussion lacks a coherent narrative.

• Line 329: "reduces the degrees of freedom of the player?" - Clarify if it means that the reaction to a stimulus in the task limits the degrees of freedom of the player's sensorimotor system.

• Lines 331-332: The sentence, “To the best of our knowledge, no other studies have investigated the influence of including decision-making on movement variability (i.e., entropy) during CoD motor tasks,” seems unsupported in the context of the specified study limitations - Lines 382-383, “Lastly, decision-making speed was not examined, which could provide valuable insights into perceptual-cognitive abilities.”

• The limitations section should be broader, particularly addressing the small sample size issue.

Reviewer #2: This study aimed to test whether reactive agility and change of direction (COD) are related or independent capacities in terms of completion time and entropy. The study is well-written, easy to read, and provides new insights into the field of field testing for soccer, considering two aspects of particular actuality in the literature of this field, such as COD and agility. This also has the merit of including for the first time (to the best of the Authors knowledge) the use of IMU systems (and the related entropy calculation) in the COD/agility assessment. The research methodology is well-structured and clear and results are presented with clarity. However, I would suggest to consider the use of some Figures instead of Tables, thus allowing the readers an immediate comprehension of the data behaviours. Discussion is clear and provides and in-depth analysis of the findings in light of the literature. Overall, I would like to congratulate the Authors for their work that certainly deserves publication.

I have only a couple of minor suggestion that I hope can contribute to complete the overall quality of the manuscript.

Line 72-73. Not clear. Ok, this is useful to introduce the concept of entropy but it seems a connection with the previous paragraph is lacking. Please rephrase or make it clearer. Consider to dedicate also more space to address this concept.

Line 313-325. This paragraph is very well-written and addresses the peculiarities of aspects related to the agility test proposed (cognitive load, more than two options, human stimuli), that contributes to a better ecological validity. However, for intellectual honesty, it is also important to highlight evidences on the importance of cognitive performance (tested using computer based cognitive tasks, i.e., with low ecological validity) for team sport performance. Please consider to dedicated space addressing this aspect. Here a couple of references on the issue to cite:

Scharfen, H.-E., Memmert, D., 2019. Measurement of cognitive functions in experts and elite athletes: A meta-analytic review. Appl. Cogn. Psychol. 33, 843–860. https://doi.org/10.1002/acp.3526

Trecroci, A., Cavaggioni, L., Rossi, A., Moriondo, A., Merati, G., Nobari, H., Ardigò, L.P., Formenti, D., 2022. Effects of speed, agility and quickness training programme on cognitive and physical performance in preadolescent soccer players. PLOS ONE 17, e0277683. https://doi.org/10.1371/journal.pone.0277683

Line 371. Have the Authors considered to test COD tasks (pre-planned and reactive) with the control of the ball? Is it possible to expect different behaviours as compared to those presented? This certainly would have imposed a more cognitive demands on the players, increasing the complexity of the task. Please elaborate.

I do believe that enlarge the possible practical applications derived from these findings might be helpful for the readers and also for practitioners working with athletes directly on the field.

Reviewer #3: General comments

The purpose of this study is interesting. However, there are several concerns and suggestions that warrant the authors' attention before reaching any decisions.

Title

I have a reservation about the use of the term "reactive" with agility. The reactive character is an inherent feature of agility

Abstract

I suggest to simply use agility and omit the term reactive

Line 44: reads awkward. Please, rephrase

Introduction

This study addressed a topic which has already been investigated. However, the authors aimed to control for entropy, a factor that hasn't been taken into account in the earlier studies. Therefore, I think that this study is based on a sound rationale and that the main purpose complement previous findings. However, the choice of female players as well as the sport discipline is not justified.

I suggest to simply use agility and delete the word "reactive" throughout the manuscript

Line 67-71: It is not that straightforward to grasp the main idea of this statement. Please reword and try to make it easier to understand. I always prefer using short and clear sentences.

Line 74: More details about linear and non-linear techniques analysis are required.

Line 79: I'm not accustomed to the concept of "entropy" but I'm looking forward to the methods section and how the authors approached this.

Line 88: I would add the appropriate literature references that guided you to such a hypothesis

Methods

Line 93: As this is a cross-sectional study, I'm concerned about the very small size, making study underpowered. I’m interested to hear from the authors about this crucial aspect.

I know that conducted a priori power analysis is not a very strong approach to justify the choice of the sample size. In other words, the process can be adapted to make it in line with the number of the participants available. This would mean that it is not "a priori" anymore but just used to somehow justify the sample size. With this in mind, could you please justify the choice of large diff (ES = 0.09) and the 95% power instead of the most common 80% power?

Line 124: "of the exist gates to avoid?"

Line 134-136: This unclear to me. Does it mean that each of the three preplanned and three reaction run need to be performed four times, meaning 12 preplanned and 12 reactive runs in total?

182: is this correct? ensuring or not ensuring?

188: I would appreciate more details related to the determination of entropy. I don't think that the current description makes it easy for someone to replicate the process. Overall, how entropy was quantified is unclear.

198: why the Kolmogorov-Smirnov and not the Shapiro-Wilk which is know as relatively more conservative?

200: The other alternative would be to log transform the data a consistently run parametric tests. It is just an idea.

205: The term interpretation was repeated twice in the same statement. Please, rephrase.

Results

224: As reliability analysis was conducted between the trials within the same session, I suggest to speak of "within session reliability"

Table 1: I'm wondering what the reason of calculating the CV expressed as %of TEM and TEM. As far as I know, the common approach is the calculate the TEM and express it as CV.

Table 2: “N=204” This value is unclear. Beneath the table you mentioned that N refers to sample but this is not clear. The sample included 17 subjects

244: I thought that this would be the other way around, that is more entropy in agility compared to CoD speed. As the ES values are negative, do you actually mean "significantly lower entropy"?

282: space is missing

Fig 4: Panel A of the figure: There are a number of outliers that could have affected the association. Have you checked that?

Discussion

312-318: These statements would better serve the rationale than the discussion. As such, I suggest to use them in the introduction.

27-331: This seems to me counter-intuitive but I may be wrong. Interested to hear further from the authors.

332: typo. should be "during"

355-359: This sentence is unclear to me.

376: There is a typo here (standardised)

Conclusion

389-392: Too long sentence and reads somehow awkward

6. PLOS authors have the option to publish the peer review history of their article (what does this mean?). If published, this will include your full peer review and any attached files.

Reviewer #1: No

Reviewer #2: No

Reviewer #3: **Yes: **Helmi Chaabene

---

## [Author Response · Author response to Decision Letter 0]

20 Nov 2023

Comments to Author:

Reviewer #1: 

We appreciate the helpful comments of Reviewer # 1. In order to clarify how we have responded, we have cut-and-pasted each comment below, responded to the comment and then explained how the comment has been accommodated in the revised manuscript. The changes in the revised manuscript are indicated in red font.

The study aimed to examine the independence of reactive agility (RA) and change of direction speed (CoD) utilizing the same movement pattern. Basic agility assessment metrics included completion time and entropy. The authors applied an ecological validity approach in measuring agility. While the paper is well-prepared, the chosen topic lacks novelty, given prior studies establishing the independence of CoD and RA (e.g., see Young, W. B., Dawson, B., & Henry, G. J. (2015). Agility and Change-of-Direction Speed are Independent Skills: Implications for Training for Agility in Invasion Sports. International Journal of Sports Science & Coaching, 10(1), 159–169). Moreover, kinematic distinctions in knee joint mechanics during running have been noted between RA and CoD tasks (Spiteri et al., 2015a; Thomas et al., 2020). Additionally, CoD plays a mediating role between sports discipline and RA (Domaradzki et al., 2021). The rationale for the study lacks persuasiveness in light of these earlier findings. If controversies exist concerning the CoD and RA relationship, the authors should emphasize them in the introduction, providing a clear justification. It is crucial to emphasize the new contributions to the existing literature, especially considering the established relationship between CoDs and RA, which is also confirmed under ecological validity conditions (Spiteri et al.)

The introduction requires further development. Some terms are poorly explained and challenging to follow. The literature review is inadequate.

We have revised the introduction to enhance the clarity of the research justification and its contribution to the existing literature. Additionally, we expanded the literature review to incorporate more sources and foundational information. All modifications made to the text are highlighted in red font within the manuscript.

In summary, while the differentiation of this concept has been studied and well-established, traditional assessment have solely relied on completion time or result variables based on the use of linear measures for the comparison. However, our research includes the exploration of the movement also using non-linear measures like entropy. This analysis allows us for understanding the evolution of human movement over time, emphasizing the exploratory nature of motion and facilitating the analysis of motor behaviour and its variations, focusing on the coordinative aspect of the execution of the action.

• Line 64: "growing number of studies" – specify the studies for clarity.

Some of the studies are cited at the end of the sentence:

“Based on this differentiation in concept, there has recently been a growing number of studies dedicated to testing whether or not COD speed and agility are independent abilities and skills, finding a positive answer to this question in most cases (i.e. better COD speed does not necessarily equate to better agility performance, and vice versa) (12–14).”

• Lines 71–73: The sentence is unclear. Provide sources and rationale.

We have modified the text and amplified the information to make it clearer:

“Notwithstanding the classical analysis of result variables provides relevant information about the outcome of the action, it remains incomplete as the underlying cause of the achieved result remains unknown. The use of non-linear techniques can fill this gap, as it takes into consideration the temporal structure and complexity of the data for the analysis (15). This approach allows for understanding the evolution of human movement over time, emphasizing the exploratory nature of motion and facilitating the analysis of motor behaviour and its variations (16,17). Thus, the use of non-linear techniques has been determined as an excellent alternative to exploring the nature of human movement and its relationship with the coordinative aspects of action providing both quantitative and qualitative insights into the behaviour of the motor system (15,18).”

• Lines 88–89: The hypothesis formulated in the paper should align with prior literature.

Comment acknowledged. We have aligned the hypothesis with prior literature and included appropriate references accordingly.

“We hypothesized that COD speed and agility are different capacities, considering the action's time (12,14) and entropy (26,27). Specifically we expect finding longer completion times and lower entropy during agility tasks (27).”

• Line 92: I suggest using the term “Participants” instead of “Subjects”.

Comment acknowledged. We have modified it in the manuscript.

• Line 97: The choice of a large effect size (0.9) for sample size calculation seems questionable. A sample size of 17 appears small to ensure measurement reliability.

Comment acknowledged. There was a mistake in the sample size description. Despite the primary focus not reliability, we wanted to detect a difference between planned COD and agility. As such, we have calculated the sample size needed to detect an ICC of 0.9 at 80% power, which confirmed 17 players were enough (14 though 16, taking into consideration a 10% dropout) (https://wnarifin.github.io/ssc/ssicc.html).

• Lines 126, 138 – The titles of the figures should be more specific.

Comment acknowledged. We have modified the titles to be more specific as follows:

“Fig 1. Graphical representation of distances, angles, and structural characteristics of the football agility task performed.”

“Fig 2. The three reaction options based on the movement of the two testers in the agility task.” 

The discussion lacks a coherent narrative.

Comment acknowledged. We have revised all the discussion to enhance its narrative coherence. Adjustments have been made in the text to ensure a clearer and more cohesive structure. You can find the modifications made to the text highlighted in red font.

• Line 329: "reduces the degrees of freedom of the player?" - Clarify if it means that the reaction to a stimulus in the task limits the degrees of freedom of the player's sensorimotor system.

Thanks for the appreciation. We have clarified it in the text as follows:

“Arguably, the inclusion of a constraint, such as the reaction to a stimulus, reduces the degrees of freedom of the player’s sensorimotor system.”

• Lines 331-332: The sentence, “To the best of our knowledge, no other studies have investigated the influence of including decision-making on movement variability (i.e., entropy) during CoD motor tasks,” seems unsupported in the context of the specified study limitations - Lines 382-383, “Lastly, decision-making speed was not examined, which could provide valuable insights into perceptual-cognitive abilities.”

We refer to the fact that entropy has never been assessed in a task involving the reaction to a stimulus, and in the limitation sections, we refer to the decision time of the action understood as the measurement of the time between stimulus presentation and the initiation of the response. We have modified both sentences to make it clearer. 

“To the best of our knowledge, no other studies have investigated the influence of including the reaction to a stimulus on movement variability (i.e., entropy) during COD motor tasks.”

“Lastly, the decision-time of the agility task understood as the time between the stimulus presentation and the initiation of the response, was not examined, which could provide valuable insights into perceptual-cognitive abilities and thus is a recommended area for future research.”.

• The limitations section should be broader, particularly addressing the small sample size issue.

Comment acknowledged. We have included this issue in the limitation section as follows:

“Additionally, the study only included female football players of a particular skill level and had a restricted sample size although sufficiently powered. Future research should aim to investigate gender comparisons, explore different skill levels, and consider expanding the sample size to attain more representative results.”

Reviewer #2: 

We appreciate the helpful comments of Reviewer # 2. In order to clarify how we have responded, we have cut-and-pasted each comment below, responded to the comment and then explained how the comment has been accommodated in the revised manuscript. The changes in the revised manuscript are indicated in red font.

This study aimed to test whether reactive agility and change of direction (COD) are related or independent capacities in terms of completion time and entropy. The study is well-written, easy to read, and provides new insights into the field of field testing for soccer, considering two aspects of particular actuality in the literature of this field, such as COD and agility. This also has the merit of including for the first time (to the best of the Authors knowledge) the use of IMU systems (and the related entropy calculation) in the COD/agility assessment. The research methodology is well-structured and clear and results are presented with clarity. However, I would suggest to consider the use of some Figures instead of Tables, thus allowing the readers an immediate comprehension of the data behaviours. Discussion is clear and provides and in-depth analysis of the findings in light of the literature. Overall, I would like to congratulate the Authors for their work that certainly deserves publication.

First, thank you for your positive feedback and valuable suggestions. We appreciate your recognition to out study’s clarity and contribution to the field. 

Also, and following your suggestion, we have incorporated a figure (The figure 4) instead of the table 2 with the basic descriptive statistical analysis to enhance data comprehension:

I have only a couple of minor suggestion that I hope can contribute to complete the overall quality of the manuscript.

Line 72-73. Not clear. Ok, this is useful to introduce the concept of entropy but it seems a connection with the previous paragraph is lacking. Please rephrase or make it clearer. Consider to dedicate also more space to address this concept.

We have rewritten this part of the introduction amplifying some of the information to make it clearer.

“Notwithstanding the classical analysis of result variables provides relevant information about the outcome of the action, it remains incomplete as the underlying cause of the achieved result remains unknown. The use of non-linear techniques can fill this gap, as it takes into consideration the temporal structure and complexity of the data for the analysis (15). This approach allows for understanding the evolution of human movement over time, emphasizing the exploratory nature of motion and facilitating the analysis of motor behaviour and its variations (16,17). Thus, the use of non-linear techniques has been determined as an excellent alternative to exploring the nature of human movement and its relationship with the coordinative aspects of action providing both quantitative and qualitative insights into the behaviour of the motor system (15,18).”

Line 313-325. This paragraph is very well-written and addresses the peculiarities of aspects related to the agility test proposed (cognitive load, more than two options, human stimuli), that contributes to a better ecological validity. However, for intellectual honesty, it is also important to highlight evidences on the importance of cognitive performance (tested using computer based cognitive tasks, i.e., with low ecological validity) for team sport performance. Please consider to dedicated space addressing this aspect. Here a couple of references on the issue to cite:

Scharfen, H.-E., Memmert, D., 2019. Measurement of cognitive functions in experts and elite athletes: A meta-analytic review. Appl. Cogn. Psychol. 33, 843–860. https://doi.org/10.1002/acp.3526

Trecroci, A., Cavaggioni, L., Rossi, A., Moriondo, A., Merati, G., Nobari, H., Ardigò, L.P., Formenti, D., 2022. Effects of speed, agility and quickness training programme on cognitive and physical performance in preadolescent soccer players. PLOS ONE 17, e0277683. https://doi.org/10.1371/journal.pone.0277683

Thanks for the recommendation. We have incorporated this concern during the paragraph following your suggestion as follows:

“In this context agility tasks containing generic stimuli such as flashing lights and arrows, although argued to provide a more challenging scenario compared to pre-planned tasks and having demonstrated their potential for enhancing athletic performance with their training and testing (44,45), still lack ecological validity.”

Line 371. Have the Authors considered to test COD tasks (pre-planned and reactive) with the control of the ball? Is it possible to expect different behaviours as compared to those presented? This certainly would have imposed a more cognitive demands on the players, increasing the complexity of the task. Please elaborate.

In a previous study we have studied the implications of adding constraints to a COD task such as the reaction to a stimulus and the ball dribbling, finding a greater reduction in speed and entropy with the inclusion of the ball and/or the reaction compared to the pre-planned and no ball option. We did not include it in this research as our objective was to compare the COD and agility capacities as they have been always studied adding the assessment of entropy. Here you have our previous research about that:

 Morral Yepes M, Gonzalo-Skok O, Fernández Valdés B, Bishop C, Tuyà S, Moras Feliu G. Assessment of movement variability and time in a football reactive agility task depending on constraints. Sport Biomech [Internet]. 2023;00(00):1–17. Available from: https://doi.org/10.1080/14763141.2023.2214533

Also, we have included it as a possibility to add in a future comparation in the limitation section following your suggestion:

“Also, it could be interesting to examine the use of different standardised sport-specific stimuli, such as reacting to a pass / opponent or including constraints that could enhance complexity of the agility task like a dribbling.”

I do believe that enlarge the possible practical applications derived from these findings might be helpful for the readers and also for practitioners working with athletes directly on the field.

Comment acknowledged. We have taken your suggestion into consideration and expanded upon the practical applications derived from our finding in the concluding section.

“In conclusion, the results of this study found significant differences, and weak to moderate correlations, between CoD speed and agility in terms of completion time and entropy, highlighting that these are two distinct, independent qualities which should be assessed and trained as such. Also, modifying task constraints is crucial to enhance athlete’s movement possibilities. Specifically, incorporating cognitive factors into a change of direction task can make it more sport-specific, potentially influencing the biomechanical and coordinative aspects of the action. In this regard, it could be interesting to initiate in a structured training program by focusing on the development of COD speed and gradually increasing complexity with the inclusion of different constraints, such as the reaction to a stimulus. This progression in the level of difficulty by the inclusion of cognitive constraints would induce changes at both coordinative and cognitive levels requiring new states of adaptation to the athlete.”

Reviewer #3: 

General comments

The purpose of this study is interesting. However, there are several concerns and suggestions that warrant the authors' attention before reaching any decisions.

We appreciate the helpful comments of Reviewer # 3. In order to clarify how we have responded, we have cut-and-pasted each comment below, responded to the comment and then explained how the comment has been accommodated in the revised manuscript. The changes in the revised manuscript are indicated in red font.

Title

I have a reservation about the use of the term "reactive" with agility. The reactive character is an inherent feature of agility.

We included the adjective “reactive” to differentiate it from COD speed, as in many studies, even currently, the term “agility” is used interchangeably for both COD speed and changes of directions in response to a stimulus. However, we agree that the term “agility” inherently implies the reaction to a stimulus, so we have removed the adjective throughout the entire text. 

Abstract

I suggest to simply use agility and omit the term reactive

Comment acknowledged. We have omitted it throughout the entire text.

Line 44: reads awkward. Please, rephrase.

Thank you for the feedback. We have revised the sentence for better clarity. The updated version now reads:

“Modifying task constraints including a reactive stimulus (i.e., cognitive factors), is essential for increasing task complexity by altering the biomechanical and coordinative aspects of the action.”

Introduction

This study addressed a topic which has already been investigated. However, the authors aimed to control for entropy, a factor that hasn't been taken into account in the earlier studies. Therefore, I think that this study is based on a sound rationale and that the main purpose complement previous findings. However, the choice of female players as well as the sport discipline is not justified.

We appreciate your acknowledgment of the study’s rationale regarding entropy control, which distinguishes it from earlier investigations.

Regarding the choice of female players and the sport discipline, we selected football due to the high frequency of agility actions in the game, playing a critical role in sport performance. 

The decision to examine female players was motivated by the growing prominence of women’s football, providing a valuable context for understanding performance dynamics. Also, choosing a female sample allow us to address the data gap in women’s sports and mitigate bias towards males in research, as emphasized by the documented lack of comprehensive data:

Cowley, E. S., Olenick, A. A., McNulty, K. L., & Ross, E. Z. (2021). “Invisible sportswomen”: the sex data gap in sport and exercise science research. Women in Sport and Physical Activity Journal, 29(2), 146-151.

Additionally, we have the opportunity to have a sample composed by high-level athletes, ensuring a robust examination of the chosen actions in highly trained population.

I suggest to simply use agility and delete the word "reactive" throughout the manuscript.

Comment acknowledged. We have omitted it throughout the entire text.

Line 67-71: It is not that straightforward to grasp the main idea of this statement. Please reword and try to make it easier to understand. I always prefer using short and clear sentences.

We have revised this paragraph to enhance clarity, and dividing it into two shorter phrases as follows:

“While insightful, the main limitation with previous research is that the conclusions were based solely on result variables as completion times, and the use of linear measures, based on the analysis of the data dispersion in relation to mean, such as standard deviation (SD) or the coefficient of variation (CV), for analysing the differences. This approach offered only quantitative information about the magnitude of the variation but overlooked the analysis of the execution and movement strategies, such as the coordination of the sporting action.”

Line 74: More details about linear and non-linear techniques analysis are required.

We have rewritten this part of the introduction amplifying some of the information to make it clearer.

“Notwithstanding the classical analysis of result variables provides relevant information about the outcome of the action, it remains incomplete as the underlying cause of the achieved result remains unknown. The use of non-linear techniques can fill this gap, as it takes into consideration the temporal structure and complexity of the data for the analysis (15). This approach allows for understanding the evolution of human movement over time, emphasizing the exploratory nature of motion and facilitating the analysis of motor behaviour and its variations (16,17). Thus, the use of non-linear techniques has been determined as an excellent alternative to exploring the nature of human movement and its relationship with the coordinative aspects of action providing both quantitative and qualitative insights into the behaviour of the motor system (15,18).”

Also, more information has been added into the introduction part trying to give more information about that.

Line 79: I'm not accustomed to the concept of "entropy" but I'm looking forward to the methods section and how the authors approached this.

We have amplified the information about non-linear measures and entropy through the introduction of the manuscript. All modifications made to the text are highlighted in red font within the manuscript. 

Also, we have tried to make clearer in the method section the formula used to calculate it.

Line 88: I would add the appropriate literature references that guided you to such a hypothesis

Comment acknowledged. We have aligned the hypothesis with prior literature and included appropriate references accordingly.

“We hypothesized that COD speed and agility are different capacities, considering the action's time (12,14) and entropy (26,27). Specifically we expect finding longer completion times and lower entropy during agility tasks (27).”

Methods

Line 93: As this is a cross-sectional study, I'm concerned about the very small size, making study underpowered. I’m interested to hear from the authors about this crucial aspect.

I know that conducted a priori power analysis is not a very strong approach to justify the choice of the sample size. In other words, the process can be adapted to make it in line with the number of the participants available. This would mean that it is not "a priori" anymore but just used to somehow justify the sample size. With this in mind, could you please justify the choice of large diff (ES = 0.09) and the 95% power instead of the most common 80% power?

Comment acknowledged. There was a mistake in the sample size description. As previously commented, we have calculated the minimum sample size to detect an ICC of 0.9 at 80% power for the reliability analysis (https://wnarifin.github.io/ssc/ssicc.html). As such, 17 players were enough for such a purpose (14 but 16, considering a 10% dropout). As the sample is compounded by highly-trained football players who are very difficult to access (Spain is currently the World Champion in females), we could have access to one team, and, consequently, the current sample size is reflective of the typical squad sizes associated with football teams. After that, we also checked through Gpower the sample size needed for 80% power, and the sample size required to detect a moderate effect size of 0.7 was 15. In addition, if you are unhappy with this approach, we can calculate the smallest effect size of interest (minimal detectable effect) through a sensitive analysis following the recommendations of Lakens (2022). Based on our sample size, effects sizes >0.74 would be considered the minimal statistical detectable effect.

Lakens, D. (2022). Sample size justification. Collabra: Psychology, 8(1), 33267.

Line 124: "of the exist gates to avoid?"

Thanks for the correction. We have modified it in the text:

“…with a line placed 2-m ahead of the exit gates to avoid slowing down before reaching the final line (Fig 1).”

Line 134-136: This unclear to me. Does it mean that each of the three preplanned and three reaction run need to be performed four times, meaning 12 preplanned and 12 reactive runs in total?

Yes, it is. We have added this information in the sentence as follows to make it clear:

“All these options resulted in six different runs (i.e., three pre-planned and three with reactions) and each player performed each run four times in a randomized order, resulting in a total of 12 pre-planned and 12 reactive trials per player.”

182: is this correct? ensuring or not ensuring?

It was not correct, we have deleted the first “no” in the sentence as follow:

“During self-calibration, three aspects were taken into account: (i) leaving the device immobile for 30 s; (ii) placing it in a flat area; and (iii) ensuring no other magnetic devices were around it (24).”

188: I would appreciate more details related to the determination of entropy. I don't think that the current description makes it easy for someone to replicate the process. Overall, how entropy was quantified is unclear.

Comment acknowledged. We have rewritten some part of the explanation trying to make it clearer as follows:

“The Sample entropy (SampEn) for each accelerometry signal, corresponding to each of the runs recorded through the IMU and extracted to an Excel file, was calculated using the equation proposed by by Richmann & Moorman (2000) (35). Then, dedicated routines programmed in Matlab® (The MathWorks, Massachusetts, USA) were used with a template length m of 2, and the tolerance criterion of 0.20 in the analyses.”

Also, here we paste the equation used in the study proposed by Richmann & Moorman (2000). However, we opted not to include it in the paper as we believe it may not be necessary given the inclusion of the reference from which it was extracted. If you find it potentially valuable, we can certainly include it in the manuscript.

 We form a vector m, X(1) to X(N-m+1) defined as:

X(i)=[x(i),x(i+1),…,X(i+m-1)] i=1,N-m+1

 We define for each I, by i=1, N-m

B_i^m (r)=1/(N-m+1) ×nº of d_m [X(i),X(j)]≤r, i ≠j 

 Again we define for each I, by i=1, N-m

A_i^m (r)=1/(N-m+1) ×nº of d_(m+1) [X(i),X(j)]≤r, i ≠j 

 Then we define:

B^m (r)=1/(N-m) ∑_(i=1)^(N-m)▒〖B_i^m (r)〗

A^m (r)=1/(N-m) ∑_(i=1)^(N-m)▒〖A_i^m (r)〗

 Finally, we calculate SampEn:

SampEn(m,r,N)=-ln⁡((A^m (r))/(B^m (r)))

198: why the Kolmogorov-Smirnov and not the Shapiro-Wilk which is know as relatively more conservative?

The decision to employ of the Kolmogorov-Smirnov test for assessing data normality was driven by our focus on discerning whether agility and COD speed represent different capacities, focusing the comparisons across run types rather than individual subjects. 

In other words, we compared the 12 pre-planned trials (4 for each direction) and the 12 reactive trials (4 for each direction) for each of the 17 players, resulting in a total of 204 data points for each type of run. Subsequently, given that each dataset contained more than 50 data points, the Kolmogorov-Smirnov test was chosen for its appropriateness in this context.

200: The other alternative would be to log transform the data a consistently run parametric tests. It is just an idea.

Thank you for the suggestion. While we have chosen not to log-transform the data for this study, we may explore this option in future investigations.

205: The term interpretation was repeated twice in the same statement. Please, rephrase.

Comment acknowledged. We have rephrased it as follows:

“The interpretation of intraclass correlation coefficient values aligned with the approach presented by Koo and Li (36)…”

Results

224: As reliability analysis was conducted between the trials within the same session, I suggest to speak of "within session reliability"

Comment acknowledged. We have included the specification in the text as follow:

“Within session reliability results are shown in table 1.”

Table 1: I'm wondering what the reason of calculating the CV expressed as %of TEM and TEM. As far as I know, the common approach is the calculate the TEM and express it as CV.

Comment acknowledged. There was a mistake. As such, we have modified it in the statistical section and table 1. 

“Within-session reliability analysis was computed using a 2-way random intraclass correlation coefficient (ICC) with an absolute agreement and 90% confidence intervals and the typical error of measurement (TEM). Furthermore, the TEM was also expressed as the coefficient of variation (CV).”

Table 2: “N=204” This value is unclear. Beneath the table you mentioned that N refers to sample but this is not clear. The sample included 17 subjects.

This value refers to the number of runs done on each type: 12 pre-planned trials (4 for each direction) and the 12 reactive trials (4 for each direction), for each of the 17 players, resulting in a total of 204 data points for each type of run. 

Following the suggestion of reviewer 2, we have changed the presentation of this data to a graph bar.

244: I thought that this would be the other way around, that is more entropy in agility compared to CoD speed. As the ES values are negative, do you actually mean "significantly lower entropy"?

Thanks for the appreciation. We have modified it in the text to make it clearer as follows:

“Significantly greater times (p<0.001, ES = 1.20,1.63) were found during agility tasks compared to COD speed task. Furthermore, a significantly lower entropy (p<0.05, ES = -0.64, -0.25) was observed during agility tasks compared to COD speed task.”

282: space is missing

Thanks for the appreciation.

Fig 4: Panel A of the figure: There are a number of outliers that could have affected the association. Have you checked that?

In our study, during data analysis, we identified and removed outlier stemming from incorrectly performed outputs. However, in response to your comment, we conducted an additional analysis by excluding the points that were furthest from the mean. The updated results show an even lower correlation (r=0.25; p<0.001). We have included below the results in a table and a scatter plot.

Discussion

312-318: These statements would better serve the rationale than the discussion. As such, I suggest to use them in the introduction.

Thank you for your feedback. We have considered your suggestion to move statements 312-318 to the introduction. While we acknowledged this perspective, we believe these statements are relevant also within the discussion. Additionally, we have incorporated another reviewer’s suggestion to this part that we find enhances the discussion. 

327-331: This seems to me counter-intuitive but I may be wrong. Interested to hear further from the authors.

We have modified the text trying to make it easier to understand as follows:

“Arguably, the inclusion of a constraint, such as the reaction to a stimulus, reduces the degrees of freedom of the player’s sensorimotor system. Consequently, movement possibilities are constrained by the need to control the external variability imposed by the environment requiring the athlete to employ more controlled motor, cognitive, and sensory skills (thus contributing to a longer completion time) than movements done in a pre-planned manner, in an attempt to reduce the complexity of the task and control the movement (5).”

In other words, players tend to adopt a more rigid and consistent movement patterns to manage environmental variability. The aim to increase stability and control their movements to enhance performance, especially in situations where environmental factors complicate the task. Here are some of the studies that have investigated and concluded this:

 Caballero C, Davids K, Heller B, Wheat J, Moreno FJ. Movement variability emerges in gait as adaptation to task constraints in dynamic environments. Gait Posture [Internet]. 2019;70:1–5. Available from: https://doi.org/10.1016/j.gaitpost.2019.02.002.

 Robalo RAM, Diniz AMFA, Fernandes O, Passos PJM. The role of variability in the control of the basketball dribble under different perceptual setups. Eur J Sport Sci [Internet]. 2020;21(4):521–30. Available from: https://doi.org/10.1080/17461391.2020.1759695

 Urbán T, Caballero C, Barbado D, Moreno FJ. Do intentionality constraints shape the relationship between motor variability and performance? PLoS One. 2019;14(4):1–14.

 Vaillancourt DE, Newell KM. Changing complexity in human behavior and physiology through aging and disease. Neurobiol Aging. 2002;23(1):1–11.

 Morral Yepes M, Gonzalo-Skok O, Fernández Valdés B, Bishop C, Tuyà S, Moras Feliu G. Assessment of movement variability and time in a football reactive agility task depending on constraints. Sport Biomech [Internet]. 2023;00(00):1–17. Available from: https://doi.org/10.1080/14763141.2023.2214533

332: typo. should be "during"

Thanks for the appreciation. We have made the change in the text as follows:

“However, previous research has examined the impact of including different constraints during tasks that require increased complexity or difficulty.”

355-359: This sentence is unclear to me.

In this sentence, we aimed to emphasize the significance of cognitive factors in agility test performance. We have revised the text to enhance clarity.

“Analysing why our study and many other have found no meaningful correlation, and significant differences between agility and COD speed tests, it’s essential to recognise that while pre-planned COD speed tasks are primarily influenced by anthropometric, technical, mechanical, physical and motor capacities (53), in agility tasks the perceptual and decision-making skills as reaction time, visual scanning, anticipation, pattern recognition, and the knowledge of situations are critical components, playing a determining role in performance (5,38,54).”

376: There is a typo here (standardised)

Thanks for the correction. We have corrected it in the text.

Conclusion

389-392: Too long sentence and reads somehow awkward.

Thanks for your feedback. We have revised the sentence and modified it in the text as follows:

“Also, modifying task constraints is crucial to enhance athlete’s movement possibilities. Specifically, incorporating cognitive factors into a change of direction task can make it more sport-specific, potentially influencing the biomechanical and coordinative aspects of the action.”

---

## [Editor Report · Decision Letter 1]

21 Nov 2023

Are change of direction speed and agility different abilities from time and coordinative perspectives?

PONE-D-23-32997R1

Dear Dr. Gonzalo-Skok,

We’re pleased to inform you that your manuscript has been judged scientifically suitable for publication and will be formally accepted for publication once it meets all outstanding technical requirements.

Kind regards,

Ersan Arslan, Ph.D.

Academic Editor

PLOS ONE
---

## [Editor Report · Acceptance letter]

28 Nov 2023

PONE-D-23-32997R1 

Are change of direction speed and agility different abilities from time and coordinative perspectives? 

Dear Dr. Gonzalo-Skok:

I'm pleased to inform you that your manuscript has been deemed suitable for publication in PLOS ONE. Congratulations! Your manuscript is now with our production department. 

Kind regards, 

on behalf of

Dr. Ersan Arslan 

Academic Editor

PLOS ONE